# Anemia in Celiac Disease: Prevalence, Associated Clinical and Laboratory Features, and Persistence after Gluten-Free Diet

**DOI:** 10.3390/jpm12101582

**Published:** 2022-09-26

**Authors:** Aurelio Seidita, Pasquale Mansueto, Stella Compagnoni, Daniele Castellucci, Maurizio Soresi, Giorgio Chiarello, Giorgia Cavallo, Gabriele De Carlo, Alessia Nigro, Marta Chiavetta, Francesca Mandreucci, Alessandra Giuliano, Rosaria Disclafani, Antonio Carroccio

**Affiliations:** 1Unit of Internal Medicine, Department of Health Promotion Sciences, Maternal and Infant Care, Internal Medicine and Medical Specialties (PROMISE), University of Palermo, 90127 Palermo, Italy; 2Unit of Internal Medicine, “V. Cervello” Hospital, Ospedali Riuniti “Villa Sofia-Cervello”, Palermo, Italy, and Department of Health Promotion Sciences, Maternal and Infant Care, Internal Medicine and Medical Specialties (PROMISE), University of Palermo, 90146 Palermo, Italy; 3Institute Zooprofilattico Sperimentale della Sicilia (IZSS), 90129 Palermo, Italy

**Keywords:** Celiac Disease, anemia, iron deficiency, gluten-free diet

## Abstract

Anemia is considered to be the most frequent extra-intestinal manifestation of Celiac Disease (CD). We assessed frequency, severity, morphologic features, and pathogenic factors of anemia in patients of the Sicilian Regional Network of Celiac Disease and attempted to identify putative pre-diet factors influencing anemia persistence. We retrospectively analyzed CD patients admitted to three centers between 2016–2020. 159 patients entered the study (129 females). More than half (54.7%) had mild-moderate, hypochromic and microcytic anemia, associated with below normal total serum iron and ferritin, indicative of iron deficiency anemia (IDA). One year after diagnosis, 134 patients were following ‘strict’ GFD. Hypochromic and microcytic anemia persisted in 46% of subjects who were anemic at diagnosis. Patients with persistent anemia had at diagnosis a higher prevalence of female gender (*p* = 0.02), lower body mass index (BMI, *p* = 0.01), higher prevalence of poly/hypermenorrhea (*p* = 0.02) and atopy (*p* = 0.04), and lower ferritin levels (*p* = 0.05) than the whole group of non-anemic ones. IDA is found in more than 50% of CD patients at diagnosis; nevertheless, in a lot of women IDA is not corrected by ‘strict’ GFD. Low BMI and poly/hypermenorrhea at diagnosis characterize this subgroup, suggesting that IDA might be due to iron loss rather than malabsorption, or to their coexistence/overlap.

## 1. Introduction

Celiac Disease (CD) has been reported in about 1% of the population [1,2,3,4], but it is often underdiagnosed because numerous patients report very few symptoms or their complete absence. Among these symptoms, the most common, historically, are diarrhea and weight loss. Currently, iron deficiency anemia (IDA) is often the presenting feature at diagnosis, being reported in over half of CD patients (including subclinical CD patients) [5,6,7,8], with a higher prevalence in adults than in children [7]. The associated histological alterations (i.e., atrophy of the duodenal mucosa) are responsible for malabsorption and multiple micronutrient deficiencies (e.g., iron, vitamin B12 and folic acid), which might be involved in the pathogenesis and morphologic features of the anemia. However, nutritional deficiencies alone cannot explain this phenomenon in all cases. In fact, anemia of chronic disease (ACD) or anemia of chronic inflammation could be responsible, especially in hospitalized patients [9].

A gluten-free diet (GFD) is the only effective treatment for resolving CD symptoms, including IDA. However, although this diet usually improves hemoglobin levels, it has been reported that 15−21% of CD patients can remain anemic, even after one or two years on a strict GFD [5,10].

The aims of this multicentric retrospective study were to evaluate the frequency, severity, morphologic features, and putative pathogenic factors contributing to anemia in patients of the “Sicilian Regional Network of Celiac Disease”, and to identify putative pre-diet factors influencing anemia persistence.

## 2. Materials and Methods

### 2.1. Study Design and Population

We retrospectively reviewed the clinical charts of a group of CD patients diagnosed between January 2016 and July 2020, in three “Hub” Centers of the Sicilian Regional Network of Celiac Disease [i.e., the Department of Internal Medicine, University Hospital of Palermo, Italy; the Department of Internal Medicine, “V. Cervello” Hospital, Palermo, Italy; and the Department of Internal Medicine, Hospital of Sciacca (Agrigento), Italy], two of which are located in the Province of Palermo and one in the southern Province of Agrigento. All the centers used a standard data collection form which was recently validated [11]. Data collection was therefore homogeneous in the various regional centers, which made it possible to evaluate the aspects of CD of greatest interest in our region. Details included demographic characteristics, family history, clinical features, the presence of intestinal and extra-intestinal symptoms/manifestations (dermatologic, musculoskeletal, ocular, oral and gynecological symptoms, neurological and psychiatric conditions, anemia), associated autoimmune diseases, CD-specific serum antibodies and duodenal histology, as requested on the Sicilian Regional Network of Celiac Disease form. Coexisting hypersensitivity conditions [multiple food hypersensitivity (MFH), systemic nickel allergic syndrome, self-reported milk intolerance (SRMI) and allergic rhino-conjunctivitis/asthma and atopic dermatitis] were also recorded.

### 2.2. Celiac Disease Diagnosis

CD was diagnosed following the criteria of the current guidelines (“four-out-of-five rule”): (1) typical intestinal and extra-intestinal signs and symptoms of CD; (2) antibody positivity (both immunoglobulin (Ig)A class anti-tTG and EmA in IgA-sufficient or IgG class anti-tTG and EmA in IgA-deficient subjects); (3) HLA-DQ2 and/or HLA-DQ8 positivity; (4) intestinal damage (proved by histology on duodenal biopsies according to the Marsh-Oberhuber classification) [12,13]; (5) clinical response to GFD (e.g., resolution of intestinal and/or extra-intestinal symptoms) [14,15].

### 2.3. Inclusion and Exclusion Criteria for CD Patients

Inclusion criteria were: (1) age over 18 years; (2) clinical and laboratory (at least a complete blood count, ferritin levels, and CD serology) follow-up after one year of GFD; and (3) at least two outpatient visits during the follow-up period.

Exclusion criteria were: (1) incomplete clinical charts; (2) no clinical or laboratory follow-up; (3) pregnancy at CD diagnosis or during follow-up; (4) diagnosis of other concomitant organic disease of the digestive system; and (5) alcohol and/or drug abuse.

### 2.4. Gluten-Free Diet Adherence Assessment

We evaluated adherence to the GFD after one year of follow-up. Patients were interviewed by experienced physicians about their clinical condition and self-reported adherence, using a validated score [16]. Only patients with an adherence score of three to four (i.e., following a strict GFD), were included in the analysis of anemia persistence.

### 2.5. Outcomes

#### 2.5.1. Primary Outcome Assessment: Prevalence and Main Features of Anemia in CD Patients

The frequency, severity, and morphologic characteristic of anemia were assessed by evaluating hemoglobin concentration (HGB), hematocrit (HCT), mean corpuscular volume (MCV), and mean corpuscular HGB (MCH). Anemia was defined as values below 12 g/dL in women and 13 g/dL in men. For the other reference values of the parameters, see Appendix A.

As they were possibly associated to the anemia, the following factors were determined: sex, age at onset, diagnostic delay, Body Mass Index (BMI), clinical presentation, weight loss, poly/hypermenorrhea, associated autoimmune diseases, and coexisting hypersensitivity conditions. The following parameters were assayed with commercial kits: total serum iron, ferritin, transferrin, erythrocyte sedimentation rate (ESR), C-reactive protein (CRP), vitamin B_12_, folic acid and thyroid-stimulating hormone (TSH). For the reference values of the parameters, see Appendix A.

In all patients suffering from IDA a validated diagnostic flow-chart was used to identify underlying causes [17].

#### 2.5.2. Secondary Outcome Assessment: Persistence of Anemia after GFD

To assess the frequency, severity, and morphologic features of anemia in the subgroup of CD patients following a strict GFD for one year, the same hematochemical parameters were evaluated (see ‘Primary Outcome Assessment’ section and Appendix A for the reference values). Moreover, the possible use of drugs to correct anemia (i.e., iron supplements) was evaluated and recorded for all the patients.

Finally, we compared the pre-diet clinical and hematochemical features of patients with persisting anemia to those of the other patients included in order to identify putative pre-GFD factors contributing to the persistence of anemia.

### 2.6. Statistical Analysis

Data were expressed as mean ± standard deviation (SD) when the distribution was Gaussian, and a Student’s *t*-test was used to evaluate differences in means between groups. Otherwise, data were expressed as median and range, and then analyzed with the Mann-Whitney U test. The χ^2^ test and Fisher’s exact test were used when appropriate. All analyses were performed using the SPSS software package (version 27.0, SPSS Inc., Chicago, IL, USA).

All subjects agreed to participate in the study. This study was performed in line with the principles of the Declaration of Helsinki. The study protocol was approved by the Ethics Committee of the University Hospital of Palermo and registered on the ClinicalTrials.gov website (registration number: NCT05172895, accessed on 29 December 2021).

## 3. Results

During the study period a total of 373 subjects were diagnosed with CD; of these, 214 were excluded on the basis of the exclusion criteria and 159 entered the present study (see Appendix A) (129 females, mean age 35.4 ± 14.7 years). Appendix A shows the demographic, clinical, histological, and serological features of the patients.

### 3.1. Prevalence and Main Features of Anemia in CD Patients

More than half (*n* = 87, 54.7%) of the total CD patients showed mild-moderate anemia [HGB (mean ± SD) 10.3 ± 1.6 g/dL], characterized by hypochromic, microcytic and anisopoikilocytosis features (data not shown).

Table 1 shows the main demographic, clinical and histological differences between the forementioned subgroups. Anemia was more commonly associated with female sex (*p* = 0.0001), a longer diagnostic delay (*p* = 0.05), and extra-intestinal symptoms (*p* = 0.0001). No other statistically significant differences were demonstrated.

Table 2 reports the main laboratory differences between anemic and non-anemic CD patients. Anemic patients presented with below normal range total serum iron, ferritin and vitamin B12, and, on the contrary, with above normal ESR. No other statistically significant differences were demonstrated.

### 3.2. Persistence of Anemia after GFD

We interviewed the whole enrolled CD population after one year of GFD, assessing their compliance to the diet with the above-mentioned adherence score. In total, 134 (84.3%) patients had a score of three to four (“strict” GFD), and these patients were therefore reassessed to identify the frequency of anemia persistence. All these patients reported either a significant improvement in symptoms compared with those at diagnosis or their complete resolution and negativization of serological CD markers.

Moreover, all these patients have assumed iron supplementation (ferrous sulphate, 595–1190 mg daily) for at least 3 months during the 1 year of GFD.

The persistence of anemia [HGB (mean ± SD) 11.0 ± 0.9 g/dL] with hypochromic and microcytic features was proved in 40 patients (46% of the subjects anemic at diagnosis), with a higher prevalence in females (*p* = 0.02). Table 3 shows the pre-GFD demographic, clinical, and hematochemical features of these patients compared to the whole group of non-anemic ones (non-anemic before GFD plus non-anemic after GFD). Patients with persistent anemia had lower BMI (*p* = 0.01) and ferritin levels (*p* = 0.05), but a higher prevalence of poly/hypermenorrhea (*p* = 0.02) and atopy (which includes allergic rhino-conjunctivitis/asthma and atopic dermatitis) (*p* = 0.04), and above normal ESR values (*p* = 0.012). No other statistically significant differences were demonstrated.

## 4. Discussion

In Western countries, IDA prevalence in adults varies according to age and sex: it is <1% in men under 50 years, 2–4% in men over 50 years, 9–20% in menstruating teenagers and young women, and 5–7% in post-menopausal women [18]. The most common pathogenetic mechanisms of IDA in adults are increased menstrual flow, occult intestinal bleeding or reduced iron absorption, as occurs in CD [19,20].

Several studies have investigated the prevalence of CD in IDA patients, finding variability across time and geographical areas [21]; however, many of them were biased by the methodologic diagnostic approach used: some studies considered only antibody positivity to CD, reporting a pooled CD prevalence of 1.4%, whereas studies based on biopsy-confirmed CD reported a prevalence of 0.7% [22]. More relevant data were reported in a metanalysis by Mahadev et al.; the authors analyzed 18 studies from several countries (including Italy), reporting a biopsy-confirmed CD prevalence in 3.2% of IDA patients. This value rose to 5.5% when only the eight studies fulfilling all the quality criteria were considered [6].

Anemia is probably the most frequent extra-intestinal manifestation of CD [23]. A recent metanalysis showed that its prevalence varies widely between studies, ranging from 12% to up to 85%, and it is more common in the female sex [22]. In our study, 54.7% of CD patients were anemic and almost all of them were women (92%, *p* = 0.0001), confirming the above-reported literature data. Of note, 70 of them also presented other extra-intestinal symptoms, much more frequently than non-anemic CD patients (80.4% vs. 43.6%, *p* = 0.0001), and a greater diagnostic delay (median 48 vs. 24 months, *p* = 0.05). This evidence is in line with other reports in the literature [24,25,26], confirming that, in CD patients, anemia is the main or the only clinical evidence for a long time, and this lack of intestinal symptoms could delay diagnosis and aggravate malabsorption, as well as increase the risk of complications, such as liver damage and neuropathy [27].

The morphologic and laboratory data of our population are typical of IDA, and both circulating and deposit levels of iron were significantly lower in the anemic than in the non-anemic CD patients. In addition, vitamin B_12_ and/or folic acid deficiencies were found in more than half (56.3%, for both) of the anemic patients. Our evidence is higher than in the literature data, which reported a prevalence of vitamin B_12_ and/or folic acid deficiency at CD diagnosis in 8–41% and 20–30% patients, respectively [28]. Morphologic features of the red blood cells, however, did not include macrocytosis; this could probably be explained by considering the absolute values of vitamin B_12_ and folic acid in the anemic CD patients, which were just slightly below the normal range, and no statistically significant differences compared with the non-anemic CD patients were found. Such deficiencies would therefore seem to be an additional factor influencing hyporegenerative anemia, which, however, is probably mainly caused by iron metabolism alterations.

Usually, IDA is associated with atrophy of the villi, becoming more marked as the histological degree of the lesions increases [29], even if it is also present in CD patients with less atrophy [30]. Of note, in our study 152 (95.6%) patients (anemic plus non-anemic) had Marsh 3 lesions and the remaining seven had Marsh 2 lesions, proving that there was no statistical difference between the anemic and non-anemic populations. Unfortunately, the low number of patients with moderate lesions did not allow a comparison to assess whether patients with lower degree lesions had equivalent or higher HGB values.

Furthermore, a chronic disease, with an associated chronic inflammatory status, could play a possible pathogenetic role in CD anemic patients; however, we did not specifically study the CD inflammatory cytokine profile (e.g., interleukin-6, and interferon-γ), which can modify erythropoiesis [31,32,33], as we analyzed only CRP and ESR. The former is known to be rarely abnormal in CD patients, either with or without anemia [5,34], while the latter is usually higher in anemic subjects “per se” because of the change in the blood to plasma ratio [35]. In our patients, we only demonstrated more above-normal values and higher absolute values of ESR in the anemic than in the non-anemic CD patients, which could simply be explained by the anemic status per se.

After one year of a strict GFD (adherence score of 3–4, disappearance of symptoms and negative CD antibodies), we reassessed anemia in our population, proving its persistence in 40 subjects (46%) [HGB (mean ± SD) 11.0 ± 0.9 g/dL]. Our results are in line with other studies: De Falco et al. reported that in 45.4% of 229 adult CD and IDA patients, anemia persisted after one year of GDF [34], and Sansotta et al. demonstrated similar results (about 30%) after one year of GFD, falling to 15% after 2 years of GFD [10]. By contrast, other studies have reported a lower prevalence of IDA persistence on GFD: recently, in a prospective study, Roldan et al. showed that anemia persisted in 20% of patients following GFD for one year and in 11% of subjects following GFD for two years [36]. However, the patients enrolled in our study are almost 10 years younger than those in Roldan’s paper (31 vs. 41 years old); this can lead to a greater number of young fertile females in our population. This evidence could suggest a relevant role of menstrual blood loose in our population, in which anemia is characterized by hypochromic, microcytic and anisopoikilocytosis features that more closely reflects IDA criteria than those of the Roldan study group, in which the anemia is normocytic and, therefore, probably due to a multifactorial etiology.

In a small study in 26 newly diagnosed CD patients, Annibale et al. reported the persistence of anemia in just 5.6% of subjects after one to two years of GFD. Interestingly, the authors proved that despite IDA resolution, 55.5% of patients had sideropenia after two years of GFD [37]. It is possible that continuing our analysis for over one year (maybe two to three years of GFD) might reveal a further reduction in IDA.

Assuming that gluten stimulation in CD patients is the main culprit behind the anemia (as reported, several pathways could be hypothesized), our results might suggest the existence of two types of CD patients: those who do not have anemia or have a typical anemia resolution on GFD, and those who remain anemic on GFD. Thus, we decided to compare these two possible subtypes to identify putative pre-diet factors influencing anemia persistence. In a Finnish study involving 163 adults with confirmed CD (68% women), 38 of them (23%) had anemia at CD diagnosis, and 10 (6%) had it after one year of GFD. Anemia was more common in women, possibly reflecting the generally higher need for iron in premenopausal women, and a lower BMI was evident in the subjects with persistent IDA [26]. Our study confirms the persistence of IDA in CD women, with a lower BMI at diagnosis (*p* = 0.02 and 0.01, respectively), suggesting that this subgroup of patients could have had a worse iron deficiency condition that was not corrected despite one year of GFD.

To our knowledge, no studies have focused on poly/hypermenorrhea as a cause of blood loss and anemia persistence in CD women. As shown in our results, there was a significant prevalence of poly/hypermenorrhea (*p* = 0.02) in the persistent IDA patients, suggesting that IDA might be due to iron loss rather than malabsorption, or to their coexistence/overlap.

We also proved a correlation between atopy and IDA persistence. The association between atopy and CD has not yet been satisfactorily established. In a Danish study, subjects diagnosed with CD had a significantly higher prevalence of IgE sensitization to food mix (*p* = 0.050), wheat (*p* = 0.014) and D. pteronyssinus (*p* = 0.014) compared with individuals without CD. They also had significantly more skin prick reactivity for D. Pteronyssinus (*p* = 0.009) compared to non-CD patients. However just nine of 2297 (0.4%) patients screened had a confirmed CD diagnosis, so no definitive conclusion can be drawn [38]. In another study, over 213 CD subjects, including only 15 women (7.0% of the whole population) reported a history of atopy [39]. Comparing anemic *vs* non-anemic CD patients after one year of GFD, we noticed that the subgroup with persisting anemia was significantly associated with a history of atopy. As the relationship between atopy and CD is still under study, it is unknown if atopy may have a role in anemia refractory to GFD.

The higher percentage of patients with low serum ferritin levels after GFD in the persistent anemia subgroup probably reflected a worse and uncorrected iron deficit. Patients’ clinical records showed that all patients have assumed iron supplementation (ferrous sulphate, 595–1190 mg daily) for at least three months during the one year of GFD. Unfortunately, since the analysis we performed was retrospective, we cannot assess whether vitamin B12 or folate were prescribed. Nevertheless, it has already been proved that GFD alone, without iron supplementation, can solve IDA [37]. Moreover, some studies have pointed out that genetic conditions might be responsible for the altered iron metabolism regulation and consequent IDA persistence in CD patients, such as the A736V TMPRSS6 polymorphism and human factor engineering (HFE) gene variants [35]. This condition, although rare, might also be present in our population.

Our study has several limitations that must be mentioned. Firstly, as it was a retrospective study, we excluded many CD patients diagnosed in our centers over the time period considered, including patients with complete clinical charts and laboratory findings (this explains the quite low number of patients included). This probably created a selection bias, with more complicated cases being evaluated, which consequently required more visits and laboratory tests. Another limitation of this study is the absence of an endoscopic re-evaluation of patients after GFD to certify the resolution or otherwise of the histological lesions (villous atrophy) that could have determined anemia persistence, as reported by some authors [6,40,41]. In fact, we did not perform an endoscopic examination, as the patients showed a significant clinical improvement and negative serology after one year of GFD. In addition, we did not perform a genetic polymorphism analysis, which might have explained some of the cases of persisting anemia.

However, our study also has some strong points. The high specialization of all the centers involved also allowed us to identify CD in subjects who would otherwise have likely been overlooked. Moreover, we included in the subgroup analysis only patients who reported a strict GFD (score three to four), thus excluding the primary cause of anemia persistence: non-adherence to GFD [5,22].

## 5. Conclusions

IDA is a frequent condition in CD patients, occurring in approximately 50% of cases; it represents the most frequent extra-intestinal CD manifestation, and is often the main or the only clinical evidence for a long time. In these cases, CD diagnosis can be delayed and malabsorption aggravated; thus, physicians should carefully exclude CD diagnosis in all IDA subjects. The treatment of this condition is strict adherence to a GFD, which can resolve, alone or with iron supplementation, the underlying iron malabsorption and deficit. However, despite correct GFD adherence, in several IDA patients HGB levels do not normalize, indicating that other factors could be (co)responsible. We proved that a large percentage (40%) of the women with persistent IDA on a strict GFD had a low BMI and presented poly/hypermenorrhea at diagnosis, suggesting that IDA might be due to iron loss rather than malabsorption, or to their coexistence/overlap. Finally, a certain role could also be attributed to a coexisting history of atopy, but further studies with a prospective design are required to clarify the relevance of all these conditions.

Regardless, these data could suggest the need for personalized diagnostic and therapeutic approaches in patients with anemia and CD based on gender, BMI, and a history of gynecological disorders.

## Figures and Tables

**Table 1 jpm-12-01582-t001:** Demographic, clinical and histological characteristics of anemic and non-anemic CD patients.

	CD * Patients without Anemia (*n* = 72) (%)	CD * Patients with Anemia (*n* = 87) (%)	*p*-Value
Sex			
Female	49 (68.1)	80 (92.0)	0.0001
Male	23 (31.9)	7 (8.0)	
Age (years) at the onset	32.6 ± 14.6	31.3 ± 14.3	NS *
(mean ± SD *)			
Diagnostic delay			
(months) [median (range)]	24 (1−336)	48 (2−732)	0.05
BMI * (mean ± SD *)	22.9 ± 4.9	23.1 ± 2.9	NS *
IBS *-like symptoms			
None	13 (18.0)	19 (21.8)	NS *
Diarrhea	41 (57.0)	42 (48.3)	NS *
Constipation	11 (15.3)	13 (15.0)	NS *
Alternating bowel movements	7 (9.7)	13 (15.0)	NS *
Dyspepsia	35 (48.6)	32 (36.8)	NS *
Extra-intestinal symptoms	38 (52.8)	70 (80.5)	0.0001
Weight loss	26 (36.1)	29 (33.3)	NS *
Poly/hypermenorrhea	13 (18.0)	26 (29.9)	NS *
Associated autoimmune diseases	21 (29.2)	34 (39.1)	NS *
MFH *	8 (11.1)	4 (4.6)	NS *
Nickel hypersensitivity	8 (11.1)	7 (8.0)	NS *
SRMI *	21 (29.2)	27 (31.0)	NS *
Atopy *	15 (20.8)	11 (12.6)	NS *
Marsh-Oberhuber classification			
2	5 (6.9)	2 (2.3)	NS *
3	67 (93.1)	85 (97.7)	NS *

* BMI = Body Mass Index; CD = Celiac Disease; IBS = Irritable Bowel Syndrome; MFH = Multiple Food Hypersensitivity; NS = Not significant; SD = Standard Deviation; SRMI = Self-Reported Milk Intolerance. Atopy includes: Allergic rhino-conjunctivitis/asthma and atopic dermatitis.

**Table 2 jpm-12-01582-t002:** HGB, iron metabolism, ESR, CRP, vitamin B12, folic acid and TSH of anemic and non-anemic CD patients.

	CD * Patients without Anemia (*n* = 72) (%)	CD * Patients with Anemia (*n* = 87) (%)	*p*-Value
HGB * (g/dL) (mean ± SD *)	13.6 ± 1.1	10.3 ± 1.6	0.05
Total serum iron (below normal value-number and %)	7 (9.7)	72 (82.8)	0.0001
Total serum iron (µg/dL) [median (range)]	75.5 (36−157)	37 (18−79)	0.0001
Ferritin (below normal value-number and %)	19 (26.4)	82 (94.2)	0.0001
Ferritin (ng/mL) [median (range)]	36 (2−284)	5 (2−44)	0.0001
ESR * (above normal value-number and %)	7 (9.7)	61 (70.1)	0.001
ESR * (mm/h) [median (range)]	16 (2−42)	31.5 (3−75)	0.003
CRP * (above normal value-number and %)	6 (8.3)	11 (12.6)	NS *
CRP* (mg/dL) [median (range)]	0.81 (0.1−7.1)	0.5 (0.06−32)	NS *
Vitamin B12 (below normal value-number and %)	0 (0.0)	49 (56.3)	0.001
Vitamin B12 (pg/mL) [median (range)]	666 (201−727)	398 (113−903)	NS *
Folic acid (below normal value-number and %)	27 (37.5)	49 (56.3)	NS *
Folic acid (ng/mL) [median (range)]	2.3 (1−3)	4.8 (0.6−11.5)	NS *
TSH * (above normal value-number and %)	21 (29.1)	16 (18.4)	NS *
TSH * (μU/mL) [median (range)]	2 (0.8−7.6)	1.6 (0.7−8.03)	NS *

* CD = Celiac Disease; CRP = C-Reactive Protein; ESR = Erythrocyte Sedimentation Rate; HGB = Hemoglobin; NS = Not significant; SD = Standard Deviation; TSH = Thyroid-Stimulating Hormone. Reference values: C-reactive protein (CRP) <5 mg/L; Erythrocyte Sedimentation Rate (ESR) 2–20 mm/h; Ferritin 15–150 ng/mL; Folic acid 3.89–26.8 mcg/L; Hemoglobin (HGB) Male 13–18 g/dL, Female 12–16 g/dL; Thyroid-Stimulating Hormone (TSH) 0.35–4.94 μU/mL; Total Serum Iron Male 65–180 µg/dL, Female 30–170 µg/dL; Vitamin B_12_ 197–890 ng/L.

**Table 3 jpm-12-01582-t003:** Pre-GFD demographic, clinical, and hematochemical features of patients with persistent anemia after one year of “strict” GFD compared to the whole non-anemic ones (non-anemic before GFD plus non-anemic after GFD).

	Whole Non-Anemic CD Patients (*n* = 94) (%)	Persisting Anemic CD Patients (*n* = 40) (%)	*p*-Value
Sex Female Male	74 (78.7) 20 (21.3)	38 (95.0) 2 (5.0)	0.02
Age (years) at diagnosis (mean ± SD *)	33.2 ± 14.2	35.8 ± 14.3	NS *
Diagnostic delay (months) [median (range)]	24 (1−456)	48 (2–660)	NS *
BMI * (mean ± SD *)	24.3 ± 4.5	22.2 ± 3.44	0.01
Dyspepsia	27 (28.7)	14 (35.0)	NS *
Extraintestinal symptoms	63 (67.0)	32 (80.0)	NS *
Weight loss	37 (39.4)	18 (45.0)	NS *
Poly/hypermenorrhea	19 (20.2)	16 (40.0)	0.02
Associated autoimmune diseases	38 (40.4)	16 (40.0)	NS *
Hashimoto thyroiditis	19 (20.2)	8 (20.0)	NS *
MFH *	9 (9.6)	3 (7.5)	NS *
Atopy *	20 (21.3)	16 (40.0)	0.04
MCV * (below normal value-number and %)	52 (55.3)	25 (62.5)	NS *
RDW * (above normal value-number and %)	13 (13.8)	10 (25.0)	NS *
Ferritin (below normal value-number and %)	59 (62.8)	35 (87.5)	0.05
ESR * (above normal value-number and %)	29 (30.8)	33 (82.5)	0.012
Vitamin B12 (below normal value-number and %)	31 (33.0)	16 (40.0)	NS *
Folic acid (below normal value-number and %)	48 (51.1)	22 (55.0)	NS *
TSH * (above normal value-number and %)	29 (30.9)	6 (15.0)	NS *
ANA * positivity (above normal value-number and %)	41 (43.6)	11 (27.5)	NS *

* ANA = Anti-Nuclear Antibodies; BMI = Body Mass Index; CD = Celiac Disease; ESR = Erythrocyte Sedimentation Rate; GFD = Gluten-Free Diet; MCV = Mean Corpuscular Volume; MFH = Multiple Food Hypersensitivity; NS = Not Significant; RDW = Red Cell Distribution Width; SD = Standard Deviation; TSH = Thyroid-Stimulating Hormone. References values: Anti-Nuclear Antibodies (ANA) negative; Erythrocyte Sedimentation Rate (ESR) 2–20 mm/h; Ferritin 15–150 ng/mL; Folic acid 3.89–26.8 mcg/L; Mean Corpuscular Volume (MCV) 80–99 fL; Red Cell Distribution Width (RDW) 11–15%; Thyroid-Stimulating Hormone (TSH) 0.35–4.94 μU/mL; Vitamin B_12_ 197–890 ng/L. Atopy includes: allergic rhino-conjunctivitis/asthma and atopic dermatitis.

## Data Availability

The data presented in this study are available on request from the corresponding author. The data are not publicly available due to restriction about patient’s privacy.

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
