# Peer review of "Anemia in Celiac Disease: Prevalence, Associated Clinical and Laboratory Features, and Persistence after Gluten-Free Diet"

_jpm, 2022, doi:10.3390/jpm12101582_

Round 1

Reviewer 1 Report

Overall very well done and important. Thank you for the opportunity to review your study. 

Minor comments-list what extra intestinal symptoms you screened for. 

Also how did you define atopy- environmental -pollen/dust etc., asthma, eczema?

Page 3 second paragraph -mia - clarify what this represents

Table 2 and table 3 list ferritin ,B12, CRP, TSH, folate in a consistent way. - If ferritin is listed in below normal values in table 2, list it the same way in table 3.  It may be that Table 3  is not properly lining up and it is hard to determine what values go with the description. Please correct. 

In the discussion section-

The  age range for your study population is almost a decade younger than those in the Roldan paper reference #36. (31 vs 41 yo). This may explain why you see more iron deficiency anemia especially in females since there would be more poly/hypermenorrhea, and this can contribute to why the iron deficiency persists in this younger female group. You should comment on this.

Page 7 Last paragraph discussion of atopy . The study used in ref 38 does not support "42.3% of CD patients self- reported allergies"  This study states that 42.3% of all the participants self reported allergy, not 42.3% of the CD subjects. Elevated CD antibodies or self reported gluten intolerance was found in 2.2% of total participants. There is no statement relating the incidence of atopy in this celiac disease population and this reference should not be used. There are other studies looking at CD and asthma/atopy but the statement that the association between atopy and celiac is still under study is fair to say. 

Author Response

Dear Reviewer

We would like to thank you for your comments on our paper. We have read with great interest your precious suggestions and have tried to modify the paper to answer all of them. As suggested by Editor, we answered point to point to all your queries and used the ‘Track change’ utility to makes all the corrections in the text easy to identify.

Best regards,

Prof. Antonio Carroccio.

Overall very well done and important. Thank you for the opportunity to review your study. 

Thank you for your appreciation.

Minor comments-list

Q1: what extra intestinal symptoms you screened for. Also how did you define atopy- environmental -pollen/dust etc., asthma, eczema?

Thank you for this observation. In our data collection form, we consider these extra-intestinal symptoms: dermatologic, musculoskeletal, ocular, oral and gynecological symptoms, neurological and psychiatric conditions, anemia. We have modified the manuscript (page 2, lines 69-70).

We modified tables 1 and 3 to better specify the conditions (allergic rhino-conjunctivitis/asthma/atopic dermatitis). which were shortly classified as ‘Atopy’. In addition, we modified the text (see page 2, line 74 and page 5, lines 180-181).

Q2: Page 3 second paragraph -mia - clarify what this represents

We apologize for this error. We corrected “mia” with “anemia” (see page 3, line 102)

Q3: Table 2 and table 3 list ferritin, B12, CRP, TSH, folate in a consistent way. - If ferritin is listed in below normal values in table 2, list it the same way in table 3.  It may be that Table 3 is not properly lining up and it is hard to determine what values go with the description. Please correct. 

Thank you, we modified the layout of tables 2 and 3, as suggested, hoping they are now easier to read. Moreover, we have uniformed the reporting of the parameters.

Q4: In the discussion section-

The age range for your study population is almost a decade younger than those in the Roldan paper reference #36. (31 vs 41 yo). This may explain why you see more iron deficiency anemia especially in females since there would be more poly/hypermenorrhea, and this can contribute to why the iron deficiency persists in this younger female group. You should comment on this.

We are glad you focused attention on this point. As you suggest, it is possible that we observed a higher prevalence of iron deficiency anemia (IDA) because of the higher percentage of females in fertile age in our study group. We modified the text according to reviewer’s indication (page 8, lines 258-264)

However, the prevalence of anemia in our population at the diagnosis of celiac disease is in line with the literature (Stefanelli G, Viscido A, Longo S, et al. Persistent Iron Deficiency Anemia in Patients with Celiac Disease Despite a Gluten-Free Diet. Nutrients 2020;12:2176. Mahadev S, Laszkowska M, Sundström J, et al. Prevalence of Celiac Disease in Patients With Iron Deficiency Anemia-A Systematic Review With Meta-analysis. Gastroenterology 2018;155:374-82).

Regarding the persistence of anemia in patients after GFD, we note that our Sicilian patients have hypochromic, microcytic anemia with anisopoikilocytosis that more closely reflects the criteria of IDA than anemia features of the Roldan study group, in which the anemia is normocytic and therefore probably due to a multifactorial etiology.

Our prevalence of persistent IDA after GFD and oral iron supplementation is in line with the literature (De Falco, L.; Tortora, R.; Imperatore, N.; et al. The role of TMPRSS6 and HFE variants in iron deficiency anemia in celiac disease. Am J Hematol 2018;93:383-393) and, as already written in the manuscript, to our knowledge, this is the first study that have focused on poly/hypermenorrhea as a cause of blood loss and anemia persistence in CD women, despite the GFD and oral iron supplementation intake.

Q5: Page 7 Last paragraph discussion of atopy. The study used in ref 38 does not support "42.3% of CD patients self- reported allergies"  This study states that 42.3% of all the participants self reported allergy, not 42.3% of the CD subjects. Elevated CD antibodies or self reported gluten intolerance was found in 2.2% of total participants. There is no statement relating the incidence of atopy in this celiac disease population and this reference should not be used. There are other studies looking at CD and asthma/atopy but the statement that the association between atopy and celiac is still under study is fair to say. 

Thank you for this observation. We apologize for this error, and we deleted the sentence and the reference from the manuscript. However, we modified the text adding other data supporting the sentence reported in page 8, lines 287-288 (‘The association between atopy and CD has not yet been satisfactorily established’) (see page 8, lines 288-293).

Reviewer 2 Report

This is a well written manuscript about anemia in Celiac Disease. The methods and design are correct. The casuistic results a bit scarce, but, as a whole, the work is correct. The manuscript is well discussed and the limitations are pointed in the discussion.

As some suggestion for improving:

-It is supposed that no specific treatment for anemia was used in these patients: this fact should be stated. Do the authors think that carential anemias should be specifically treated in addition to GFD?

-What are the best markers to anemia diagnosis (or suspicion) and follow up in Celiac Disease?

Page 3, line 100: "mia" instead of "anemia".

Author Response

Dear Reviewer

We would like to thank you for your comments on our paper. We have read with great interest your precious suggestions and have tried to modify the paper to answer all of them. As suggested by Editor, we answered point to point to all your queries and used the ‘Track change’ utility to makes all the corrections in the text easy to identify.

Best regards,

Prof. Antonio Carroccio.

This is a well written manuscript about anemia in Celiac Disease. The methods and design are correct. The casuistic results a bit scarce, but, as a whole, the work is correct. The manuscript is well discussed and the limitations are pointed in the discussion.

Thank you for your appreciation.

As some suggestion for improving:

Q1: -It is supposed that no specific treatment for anemia was used in these patients: this fact should be stated. Do the authors think that carential anemias should be specifically treated in addition to GFD?

Thank you for your observation. We have already specified treatment for anemia in the discussion (page 8, lines 300-304): from the review of our patient’s charts all patients with IDA have assumed iron supplementation (ferrous sulphate, 595-1190 mg daily) for at least 3 months during the 1 year of GFD.

Moreover, we modified both ‘Materials and Methods’ (see page 2, lines 117-118) and results (page 5, lines 172-173), reporting this data.

Q2: -What are the best markers to anemia diagnosis (or suspicion) and follow up in Celiac Disease?

Unfortunately, no markers have been identified in CD patients that could suggest a sideropenic anemia in advance, or that are useful during the follow-up.  In our experience, ferritin deficiency often indicates those cases who will become anemic. However, we have no prospective data to confirm this clinical opinion and, to our knowledge, there are no specific data in the literature, even if a recent metanalysis reported that “… blood ferritin concentration is reasonably sensitive and a very specific test for iron deficiency in people presenting for medical care”.

[Garcia-Casal MN, Pasricha SR, Martinez RX, et al. Serum or plasma ferritin concentration as an index of iron deficiency and overload. Cochrane Database Syst Rev 202;5:CD011817]

Q3: Page 3, line 100: "mia" instead of "anemia".

We apologize for this error. We corrected “mia” with “anemia” (see page 3, line 101).
